# Characterization of the complete mitochondrial genome of *Orthaga olivacea* Warre (Lepidoptera Pyralidae) and comparison with other Lepidopteran insects

**Liangli Yang**[1☯]**, Junjun Dai**[2☯]**, Qiuping Gao**[1]**, Guozhen Yuan**[1]**, Jiang Liu**[1]**, Yu Sun**[1]**, Yuxuan Sun**[1]**, Lei Wang**[1]**, Cen Qian**[1]**, Baojian Zhu**[1]**, Chaoliang Liu**[1]**, Guoqing Wei**[1]*

**1** School of Life Sciences, Anhui Agricultural University, Hefei, P. R. China, **2** Sericultural Research Institute, Anhui Academy of Agricultural Sciences, Hefei, P. R. China

☯ These authors contributed equally to this work.
* weiguoqing@ahau.edu.cn

## Abstract

*Orthaga olivacea* Warre (Lepidoptera: Pyralidae) is an important agricultural pest of camphor trees (*Cinnamomum camphora*). To further supplement the known genome-level features of related species, the complete mitochondrial genome of *Orthaga olivacea* is amplified, sequenced, annotated, analyzed, and compared with 58 other species of Lepidopteran. The complete sequence is 15,174 bp, containing 13 protein-coding genes (PCGs), 22 transfer RNA (tRNA) genes, 2 ribosomal RNA (rRNA) genes, and a putative control region. Base composition is biased toward adenine and thymine (79.02% A+T) and A+T skew are slightly negative. Twelve of the 13 PCGs use typical ATN start codons. The exception is cytochrome oxidase 1 (*cox1*) that utilizes a CGA initiation codon. Nine PCGs have standard termination codon (TAA); others have incomplete stop codons, a single T or TA nucleotide. All the tRNA genes have the typical clover-leaf secondary structure, except for *trnS*(AGN), in which dihydrouridine (DHU) arm fails to form a stable stem-loop structure. The A+T-rich region (293 bp) contains a typical Lepidopter motifs 'ATAGA' followed by a 17 bp poly-T stretch, and a microsatellite-like (AT)$_{13}$ repeat. Codon usage analysis revealed that *Asn*, *Ile*, *Leu2*, *Lys*, *Tyr* and *Phe* were the most frequently used amino acids, while *Cys* was the least utilized. Phylogenetic analysis suggested that among sequenced lepidopteran mitochondrial genomes, *Orthaga olivacea* Warre was most closely related to *Hypsopygia regina*, and confirmed that *Orthaga olivacea* Warre belongs to the Pyralidae family.

## Introduction

The insect mitochondrial DNA (mtDNA) is a closed-circular molecule ranging in size from 14,000 to 19,000 bp [1]. It generally contains 37 genes, of which seven are NADH dehydrogenase subunits (*nad1-nad6* and *nad4L*), three cytochrome C oxidase subunits (*cox1-cox3*), two ATPase subunits (*atp6* and *atp8*), one cytochrome b (*cytb*) subunit, two ribosomal RNAs (*rrnL*

**Data Availability Statement:** All sequences date files are available from the GenBank at NCBI database (accession number mitochondrion MN078362).

**Funding:** GQW was supported by the grant from the National Natural Science Foundation of China (31472147) the earmarked fund for Anhui International Joint Research and Development Center of Sericulture Resources Utilization (2017R0101). The funders had no role in study design, data collection and analysis, decision to publish, or preparation of the manuscript.

**Competing interests:** The authors have declared that no competing interests exist.

and *rrnS*), and 22 transfer RNAs (tRNA) [2, 3], and a variable length A+T-rich region, the largest noncoding sequence that modulates transcription and replication [4, 5, 6]. Whole mitochondrial genomes are a useful data source for several research areas [7, 8], such as evolutionary genomics [9, 10] and comparative molecular evolution [11, 12], phylogeography [13], and population genetics [14].

The Lepidoptera (butterflies and moths) comprises over 160,000 described species, classified into 45–48 superfamilies and is cosmopolitan in distribution [15]. Pyralidae is one of the largest families in Lepidoptera, including over 25,000 species and some of pyralids are important agricultural pests, such as *Ostrinia nubilalis* and *Cnaphalocrocis medinalis*, whose complete mitogenomes had been sequenced [16–18]. Despite their diversity and great importance as pests of agricultural and forestry plants, they are also valuable for pollinating plants of economic importance. Most species in the family Pyralidae do not yet have sequenced mitogenomes.

*Orthaga olivacea* Warre (Lepidoptera: Pyralidae) is a notorious pest, widely distributed in East China. The larvae feed on *Cinnamomum camphora* leaves and cause considerable economic losses. Farmers apply chemical prevention and removal strategies to combat this pest species particularly during larval and pupa life stages [19]. However, overlapping generations and irregularity of abundance in the field from May to October make it very difficult to control [19]. Previous studies have investigated the host preference, distribution and morphological characteristics of *Orthaga olivacea* Warre, and the control of it by bio-pesticide has been investigated [20, 21]. However, the use of pesticides is harmful to the environment. Therefore, it is necessary to find new strategies to prevent this pest. In this study we sequenced the complete mitogenome of *Orthaga olivacea* Warre, and compared it with other insect species, especially with the members of Pyralidae species. Phylogenetic relationships among lepidopteran superfamilies were reconstructed using the nucleotide sequences from the 13 PCGs to test the position of *Orthaga olivacea* within Pyralidae. The study of mitogenomes of *Orthaga olivacea* can provide fundamental information for mitogenome architecture, phylogeography, future phylogenetic analyses of Pyralidae, and biological control of pests.

## Materials and methods

### Sample collection and DNA isolation

*Orthaga olivacea* Warre, larvae (the larvae are about 22–30 mm long, brown, reddish-brown on the head and anterior thoracic plate, and have a brown wide band on the back of the body, with two yellow-brown lines on each side.) were collected from the camphor trees on the campus of Anhui Agricultural University (Hefei, China). Specimens were preserved with 100% ethanol and stored at -80˚C. This insect is not an endangered or protected species. Total genomic DNA was extracted from the larvae using the Aidlab Genomic DNA Extraction Kit (Aidlab Co., Beijing, China) according to the manufacturer's instructions. Extracted DNA quality was assessed by 1% agarose (w/v) gel electrophoresis.

### Amplification and sequencing

Thirteen pairs of conserved primers were designed from the mitogenomes of previously sequenced Pyralidae species (synthesized by BGI Tech Co., Shenzhen, China) (Table 1). All PCRs were performed in 50 μL reaction volumes; 34.75 μL sterilized distilled water, 5 μL 5 × Taq buffer ($Mg^{2+}$ plus), 4 μL dNTPs (2.5 mM), 2 μL genomic DNA, 2 μL of each primer (10 μM) and 0.25 μL (1.25 unit) Taq polymerase (TaKaRa Co., Dalian, China). A two-step PCR was performed under the following conditions: initial denaturation at 94˚C for 5 min followed by 35 cycles of 30s at 94˚C, annealing 2–3 min (depending on putative length of the

**Table 1. Details of the primers used to amplify the mitogenome of *O. olivacea* Warre.**

| Primer pair | Primer sequence (5' -3') |
| --- | --- |
| F1 | TAAAAATAAGCTAAATTTAAGCTT |
| R1 | TATTAAAATTGCAAATTTTAAGGA |
| F2 | AAACTAATAATCTTCAAAATTAT |
| R2 | AAAATAATTTGTTCTATTAAAG |
| F3 | ATTCTATATTTCTTGAAATATTAT |
| R3 | CATAAATTATAAATCTTAATCATA |
| F4 | TGAAAATGATAAGTAATTTATTT |
| R4 | AATATTAATGGAATTTAACCACTA |
| F5 | TAAGCTGCTAACTTAATTTTTAGT |
| R5 | CCTGTTTCAGCTTTAGTTCATTC |
| F6 | CCTAATTGTCTTAAAGTAGATAA |
| R6 | TGCTTATTCTTCTGTAGCTCATAT |
| F7 | TAATGTATAATCTTCGTCTATGTAA |
| R7 | ATCAATAATCTCCAAAATTATTAT |
| F8 | ACTTTAAAAACTTCAAAGAAAAA |
| R8 | TCATAATAAATTCCTCGTCCAATAT |
| F9 | GTAAATTATGGTTGATTAATTCG |
| R9 | TGATCTTCAAATTCTAATTATGC |
| F10 | CCGAAACTAACTCTCTCTCACCT |
| R10 | CTTACATGATCTGAGTTCAAACCG |
| F11 | CGTTCTAATAAAGTTAAATAAGCA |
| R11 | AATATGTACATATTGCCCGTCGCT |
| F12 | TCTAGAAACACTTTCCAGTACCTC |
| R12 | AATTTTAAATTATTAGGTGAAATT |
| F13 | TAATAGGGTATCTAATCCTAGTT |
| R13 | ACTTAATTTATCCTATCAGAATAA |

fragments) at 51–58˚C (depending on primer combination) and a final extension step of 72˚C for 10 min.

PCR amplicons were analyzed on 1.0% agarose gel electrophoresis, and purified using a gel extraction kit (CWBIO Co., Beijing, China). Purified fragments were ligated into the T-vector (TaKaRa Co., Dalian, China) and transformed into *Escherichia coli* DH5α. Positive recombinant colonies with insert DNA were sequenced in both directions and at least three times by Invitrogen Co. Ltd. (Shanghai, China).

## Sequence annotation

The complete mtDNA sequence was assembly using the DNAStar package (DNAStar Inc. Madison, USA) and sequence annotation was performed using the blast tools from NCBI (http://blast.ncbi.nlm.nih.gov/Blast). The sequences were submitted to GenBank at NCBI under the accession number MN078362. The tRNA genes were identified using the tRNAscan-Se program software available online at http://lowelab.ucsc.edu/tRNAscan-SE/, and visually identify sequences using the alignment with the appropriate anticodons capable of folding into the typical clover-leaf structure [22]. PCGs were initially identified by sequence identity with Pyralidae species and aligned with the other lepidopteran using ClustalX version 2.0 [23]. Nucleotide sequences of the PCGs were translated into their putative amino acids based on the invertebrate mtDNA genetic code. Composition skew was performed according to the

formulas AT skew = [A−T]/[A+T], GC skew = [G−C]/[G+C]) [24]. Relative Synonymous Codon Usage (RSCU) values were calculated in MEGA 6.0 [25]. Tandem repeats in the A+T-rich region were predicted using the Tandem Repeats Finder program (http://tandem.bu.edu/trf/trf.html) [26].

### Phylogenetic analysis

To reconstruct the phylogenetic relationships of Lepidoptera, 58 lepidopteran mitogenomes (Table 2) representing seven lepidopteran superfamilies (Bombycoidea, Noctuoidea, Geometroidea, Pyraloidea, Tortricoidea, Papilionoidea and Yponomeutoidea) were used. The mitogenomes of *Limnephilus hyalinus* (NC_044710.1) [27], *Locusta migratoria* (NC_001712.1) [28], and *Drosophila yakuba* (NC_001322) [29] were used as outgroups. The 13 PCGs concatenated nucleotide sequences of these lepidopterans were initially aligned using ClustalX version 2.0. Phylogenetic analysis was performed using Maximum Likelihood (ML) method with the MEGA 6.0 program. This method was used to infer phylogenetic trees with 1000 bootstrap replicates.

## Results and discussion

### Genomic structure, organization and composition

The complete mitogenome of *Orthaga olivacea* Warre is a circular molecule with 15,174 base pairs (bp) in size (Fig 1). This is comparable to the mitogenome sizes documented for other sequenced lepidopterans which range from 14,535 bp in *Ostrinia nubilalis* to 16,179 bp in *Plutella xylostella*, and it is similar to *Lista haraldusalis* (15213) (Table 2). The *Orthaga olivacea* Warre mitogenome is identical to that of other lepidopterans in terms of gene organization, including all 13 PCGs (*cox1–3*, *nad1–6*, *nad4L*, *cytb*, *atp6* and *atp8*), 22 tRNA genes, two ribosomal RNAs (*rrnS* and *rrnL*), and the important non-coding region also known as "A+T-rich region" [70, 71] (Fig 1; Table 3). Variety in non-coding regions is the primarily reason for size differences across Lepidoptera mitochondrial genomes. Nucleotide composition revealed that the most common base is T = 6249 (41.18%) and the least common base is G = 1249 (8.23%) and AT skew [72] (As to Ts) is slightly negative (−0.042). This trend has also been reported from *Manduca sexta* (−0.005) [34], *Ctenoplusia agnata* (−0.023) [39], *Paracymoriza distinctalis* (−0.002) [46], and *Lista haraldusalis* (−0.007) [57]. In addition, the GC skew (Gs to Cs) is also negative (−0.215). Base composition of the *Orthaga olivacea* Warre mitogenome is A+T rich (79.02% A+T content and 20.98% G+C content). Highly A+T biased mitogenomes have been previously sequenced from lepidopterans (ranging from 77.8% in *Rondotia menciana* to 81.94% in *Cnaphalocrocis medinalis*) [17, 31], (Table 4). Nucleotide skew is negative, similar to the mitogenome of other lepidopterans, such as *M. sexta* (-0.005 and -0.181) [33] and *C.medinalis* (-0.030 and -0.175) [17] (Table 4).

### Protein-coding genes

The concatenated protein-coding genes are 11,147 bp in length, accounting for approximately 73.46% of the mitogenome. All PCGs are initiated by typical ATN start codons, except *cox1*, which is initiated by CGA (Table 3). The use of a non-canonical start codon for this gene is common across lepidopterans [17, 37, 73, 74], and *cox1* transcripts do not overlap with the upstream tRNA, as has been proposed for several insect species [75]. Annotation of *cox1* start codon can be justifiably conducted on the basis of comparative amino acid alignments, aiming to identify conserved sites downstream of the flanking tRNA, and there is thus no justification for continued speculation about polynucleotide start codon [76].

**Table 2. Details of the lepidopteran mitogenomes used in this study.**

| Superfamily | Family | Species | Size (bp) | GenBank accession no. | Reference |
|---|---|---|---|---|---|
| Bombycoidea | Bombycidae | *Bombyx mandarina* | 15,682 | AY301620 | [30] |
| | | *Bombyx mori* | 15,643 | NC_002355 | Direct submission |
| | | *Rondotia menciana* | 15,301 | KC881286.1 | [31] |
| | Saturniidae | *Antheraea pernyi* | 15,566 | AY242996 | [32] |
| | | *Antheraea yamamai* | 15,338 | NC_012739 | [33] |
| | Sphingidae | *Manduca sexta* | 15,516 | NC_010266 | [34] |
| | | *Sphinx morio* | 15299 | KC470083.1 | [35] |
| Noctuoidea | Lymantriidae | *Lymantria dispar* | 15,569 | NC_012893 | Unpublished |
| | | *Euproctis pseudoconspersa* | 15461 | KJ716847.1 | [36] |
| | Erebidae | *Amata formosae* | 15,463 | KC513737 | [37] |
| | Notodontidae | *Ochrogaster lunifer* | 15,593 | NC_011128 | [38] |
| | Noctuidae | *Ctenoplusia agnata* | 15261 | KC414791.1 | [39] |
| | | *Agrotis ipsilon* | 15,377 | KF163965 | [40] |
| | Nolidae | *Gabala argentata* | 15,337 | KJ410747 | [41] |
| Geometroidea | Geometridae | *Apocheima cinerarium* | 15,722 | KF836545 | [42] |
| | | *Biston thibetaria* | 15,484 | KJ670146.1 | Unpublished |
| Pyraloidea | Crambidae | *Chilo suppressalis* | 15,395 | NC_015612 | [43] |
| | | *Diatraea saccharalis* | 15,490 | NC_013274 | [44] |
| | | *Ostrinia furnacalis* | 14,536 | NC_003368 | [45] |
| | | *Ostrinia nubilalis* | 14,535 | NC_003367.1 | [45] |
| | | *Cnaphalocrocis medinalis* | 15388 | NC_015985 | [43] |
| | | *Paracymoriza distinctalis* | 15354 | KF859965.1 | [46] |
| | | *Tyspanodes hypsalis* | 15329 | NC_025569 | [47] |
| | | *Paracymoriza prodigalis* | 15,326 | NC_020094.1 | [48] |
| | | *Elophila interruptalis* | 15,351 | NC_021756.1 | [49] |
| | | *Pseudargyria interruptella* | 15.231 | NC_029751.1 | Direct submission |
| | | *Chilo auricilius* | 15,367 | NC_024644.1 | [50] |
| | | *Chilo sacchariphagus* | 15,378 | NC_029716.1 | Direct submission |
| | | *Evergestis junctalis* | 15,438 | NC_030509.1 | Direct submission |
| | | *Nomophila noctuella* | 15,309 | NC_025764.1 | [51] |
| | | *Tyspanodes striata* | 15,255 | NC_030510.1 | Direct submission |
| | | *Glyphodes quadrimaculalis* | 15,255 | NC_022699.1 | [52] |
| | | *Spoladea recurvalis* | 15,273 | NC_027443.1 | [53] |
| | | *Dichocrosis punctiferalis* | 15,355 | NC_021389.1 | [54] |
| | | *Glyphodes pyloalis* | 14,960 | NC_025933.1 | Unpublished |
| | | *Maruca vitrata* | 15,385 | NC_024099.1 | Unpublished |
| | | *Maruca testulalis* | 15,110 | NC_024283.1 | [55] |
| | | *Haritalodes derogat* | 15,253 | NC_029202.1 | Unpublished |
| | | *Pycnarmon lactiferalis* | 15,219 | NC_033540.1 | [56] |
| | | *Loxostege sticticalis* | 15,218 | NC_027174.1 | Unpublished |
| | Pyralidae | ***Orthaga olivacea* Warre** | | | **This study** |
| | | *Lista haraldusalis* | 15213 | NC_024535 | [57] |
| | | *Galleria mellonella* | 15320 | KT750964 | Unpublished |
| | | *Corcyra cephalonica* | 15,273 | NC_016866.1 | [58] |
| | | *Amyelois transitella* | 15,205 | NC_028443.1 | [59] |
| | | *Plodia interpunctella* | 15,264 | NC_027961.1 | Unpublished |
| | | *Ephestia kuehniella* | 15,295 | NC_022476.1 | Direct submission |
| | | *Meroptera pravella* | 15,260 | NC_035242.1 | [60] |
| | | *Hypsopygia regina* | 15,212 | NC_030508.1 | Direct submission |
| | | *Endotricha consocia* | 15,201 | NC_037501.1 | [61] |

*(Continued)*

**Table 2.** (Continued)

| Superfamily | Family | Species | Size (bp) | GenBank accession no. | Reference |
|---|---|---|---|---|---|
| | | *Euzophera pyriella* | 15,184 | NC_037175.1 | [62] |
| Tortricoidea | Tortricidae | *Grapholita molesta* | 15,717 | NC_014806 | [63] |
| | | *Spilonota lechriaspis* | 15,368 | NC_014294 | [64] |
| Papilionoidea | Papilionidae | *Luehdorfia taibai* | 15,553 | KC952673 | [65] |
| | | *Teinopalpus aureus* | 15,242 | NC_014398 | Unpublished |
| | | *Apatura ilia* | 15,242 | NC_016062 | [66] |
| | | *Apatura metis* | 15,236 | NC_015537 | [67] |
| Yponomeutoidea | Plutellidae | *Plutella xylostella* | 16,179 | JF911819 | [68] |
| | Lyonetiidae | *Leucoptera malifoliella* | 15,646 | NC_018547 | [69] |

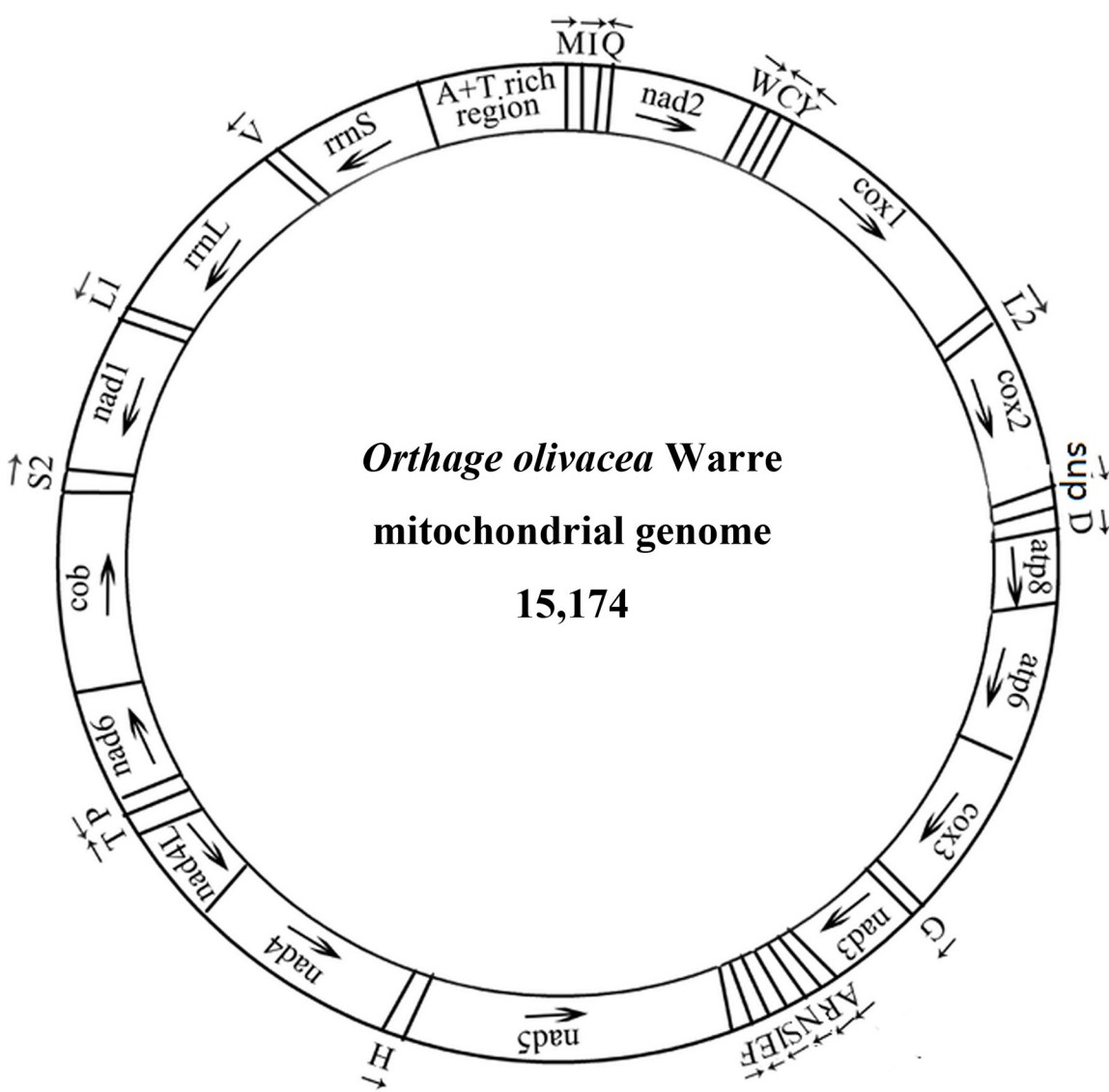

**Fig 1. Map of the mitogenome of *O. olivacea* Warre.** Labeling tRNA genes according to the IUPAC-IUB single-letter amino acids: *cox1*, *cox2* and *cox3* present the three subunits of cytochrome c oxidase; *cob* present cytochrome b; *nad1-nad6* constitutes NADH dehydrogenase; *rrnL* and *rrnS* refer to ribosomal RNAs. Genes named above the bar are located on major strand, while the others are located on minor strand. Anti-clockwise rRNA or PCGs genes are located on L strand and others are located on H strand.

**Table 3. Summary results for characteristics of the mitogenome of *Orthaga olivacea* Warre.**

| Gene | Location | Direction | Size | Intergenic Nucleotides | Start codon | Stop codon |
|------|----------|-----------|------|------------------------|-------------|------------|
| tRNA-Met | 1–67 | F | 67 | 1 | — | — |
| tRNA-Ile | 69–132 | F | 64 | -3 | — | — |
| tRNA-Gln | 130–198 | R | 69 | 52 | — | — |
| ND2 | 251–1264 | F | 1014 | 0 | ATT | TAA |
| tRNA-Trp | 1265–1332 | F | 68 | -8 | — | — |
| tRNA-Cys | 1325–1394 | R | 70 | 4 | — | — |
| tRNA-Tyr | 1399–1464 | R | 66 | 3 | — | — |
| COX1 | 1468–2973 | F | 1506 | 0 | CGA | TAA |
| tRNA-Leu1 | 2974–3040 | F | 67 | 0 | — | |
| COX2 | 3041–3712 | F | 672 | 0 | ATT | TAA |
| tRNA-Sup | 3713–3781 | F | 69 | 4 | — | — |
| tRNA-Asp | 3786–3853 | F | 68 | 0 | — | — |
| ATP8 | 3854–4015 | F | 162 | -7 | ATC | TAA |
| ATP6 | 4009–4689 | F | 681 | -1 | ATG | TAA |
| COX3 | 4689–5478 | F | 790 | 2 | ATG | T |
| tRNA-Gly | 5481–5548 | F | 68 | 0 | — | — |
| ND3 | 5549–5902 | F | 354 | 12 | ATT | TAA |
| tRNA-Ala | 5915–5980 | F | 66 | 0 | — | — |
| tRNA-Arg | 5981–6044 | F | 64 | 2 | — | — |
| tRNA-Asn | 6047–6112 | F | 66 | 3 | — | — |
| tRNA-Ser1 | 6116–6168 | F | 53 | 19 | — | — |
| tRNA-Glu | 6188–6253 | F | 66 | -2 | — | — |
| tRNA-Phe | 6252–6318 | R | 67 | 0 | — | — |
| ND5 | 6319–8052 | R | 1734 | 0 | ATT | TAA |
| tRNA-His | 8053–8118 | R | 66 | 0 | — | — |
| ND4 | 8119–9455 | R | 1337 | 0 | ATA | TA |
| ND4L | 9456–9746 | R | 291 | 2 | ATG | TAA |
| tRNA-Thr | 9749–9812 | F | 64 | 0 | — | — |
| tRNA-Pro | 9813–9877 | R | 65 | 0 | — | — |
| ND6 | 9878–10398 | F | 521 | 9 | ATA | TAA |
| CYTB | 10408–11566 | F | 1159 | -2 | ATG | T |
| tRNA-Ser2 | 11565–11631 | F | 67 | 20 | — | — |
| ND1 | 11652–12577 | R | 926 | 1 | ATG | TA |
| tRNA-Leu2 | 12579–12648 | R | 70 | 0 | — | — |
| rRNA-16s | 12649–14032 | R | 1384 | 0 | — | — |
| tRNA-Val | 14033–14096 | R | 64 | 0 | — | — |
| rRNA-12s | 14097–14881 | R | 785 | 0 | — | — |
| A-T-rich region | 14882–15174 | F | 293 | | — | — |

Nine PCGs have canonical termination codons TAA or TAG, while four have incomplete termination codons single T (*cox3* and *cytb*) or TA (*nad4* and *nad1*) (Table 3). Incomplete stop codons have been observed in most other lepidopteran mitogenomes and are common across mitogenomes [77]. It has been proposed that polycistronic pre-mRNA transcripts are processed by endonucleases, cleaving between tRNAs, and that polyadenylation of adjacent PCGs produces functional stop-codons from the partial termination codons such as a single T [78].

Complete mitogenome sequences of several lepidopterans were evaluated for codon usage. These species belonged to seven superfamilies (three species belonging to Pyraloidea, two

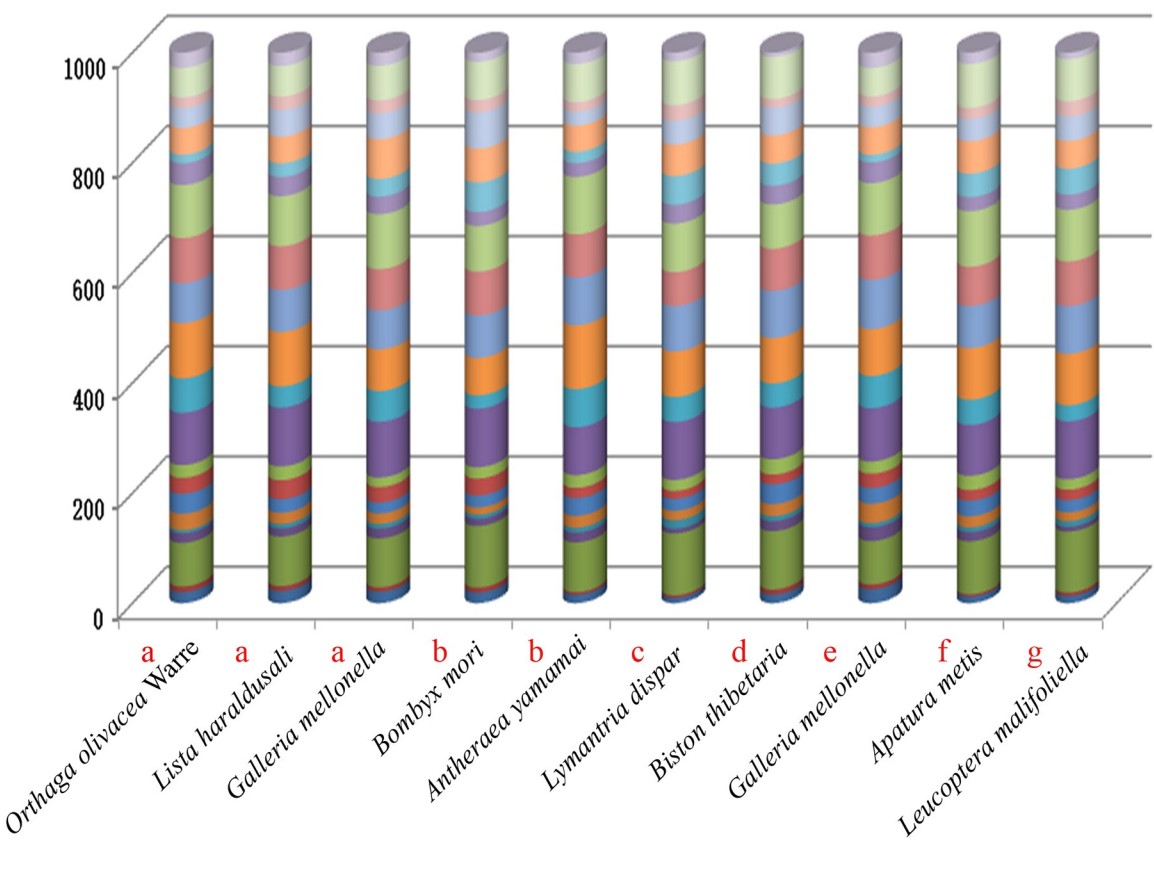

**Fig 2. Codon usage patterns of *O. olivacea* Warre mitochondrial genome compared with other species of the Lepidoptera.** The lowercase letters above species name (a, b, c, d, e, f and g) indicate the superfamily which the species belong to (a: *Pyraloidea*, b: *Bombycoidea*, c: *Noctuoidea*, d: *Geometroidea*, e: *Tortricoidea*, f: *Papilionoidea*, g: *Yponomeutoidea*).

species belonging to Bombycoidea, and one from each Noctuoidea, Geometroidea, Tortricoidea, Papilionoidea and Yponomeutoidea) (Fig 2). The analysis of codon usage showed that *Asn, Ile, Leu2, Lys, Tyr* and *Phe* were the amino acids with high relative usage frequency, while *Arg* was the least used amino acid. Three species of Geometroidea have consistent codon distributions in and each amino acid has equal content in them (Fig 3). The least used codons are those with high G and C, possibly due to high AT skew in lepidoptera PCGs [37, 79], for instance, *L. haraldusalis, G. mellonella, B. mori, B. thibetaria, and L. malifoliella* species all lack GCT codons, while *G. molesta* lacks CGT codons. However, in the present study all of these codons were observed in the mitogenome of *Orthaga olivacea* Warre (Fig 4) like that of *A. yamamai, L. dispar* and *A. metis* species [33, 67].

## Transfer and ribosomal RNA genes

*Orthaga olivacea* Warre mitogenome has 22 tRNA genes, ranging in size from 53 bp (*tRNA-Ser1*) to 70 bp (*tRNA^Cys* and *tRNA^Leu*). TRNAs show high A+T content (80.17%) and negative AT-skew (−0.015). All the tRNAs display typical cloverleaf secondary structures, except *trnS^AGN* which is missing a stable dihydrouridine (DHU) arm (Fig 5); this phenomenon is common across insects [17, 80, 81].

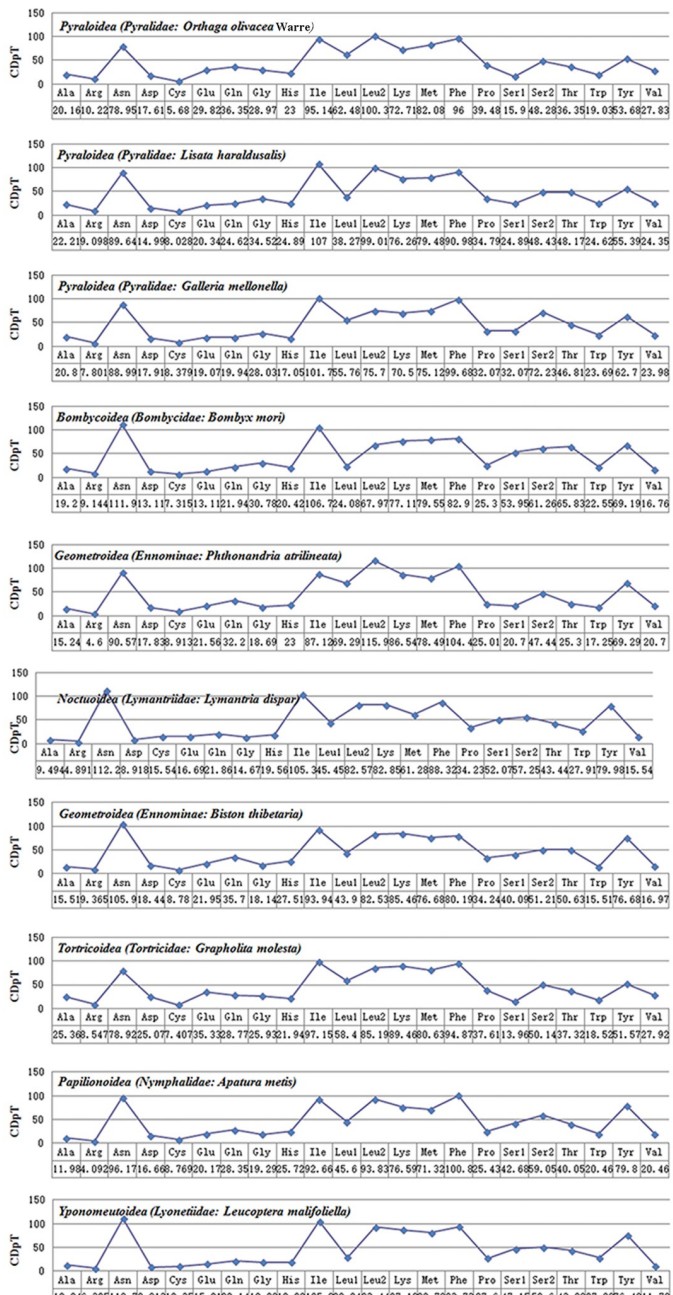

**Fig 3. Codon distribution of *O. olivacea* Warre compared with other species of the Lepidoptera.** CDspT = codons per thousand codons.

The rRNAs showed higher A+T content (84.00%) in comparison to the PCGs and tRNAs; this value falls within the range of sequenced insects (Table 4).

## Overlapping and intergenic spacer regions

Six overlapping sequences with a total length of 23 bp were identified in the *Orthaga olivacea* Warre mitogenome. These sequences varied in length from 1 to 8 bp, and between $tRNA^{Trp}$

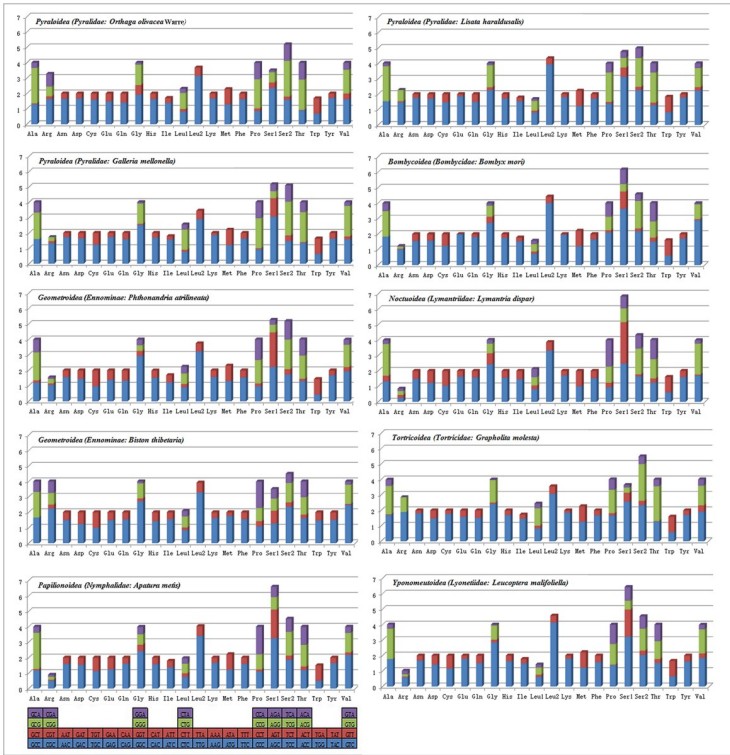

**Fig 4. The Relative Synonymous Codon Usage (RSCU) of the eight superfamilies mitochondrial genome of Lepidoptera.** Codon family is displayed on the X axis. Codons which are not present in mitochondrial genomes are indicated above.

and *tRNA^Cys^* with the biggest overlapping region (8 bp). The overlapping region located between *atp8* and *atp6* was 7 bp, 3 bp between *tRNA^Ile^* and *tRNA^Gln^*, while the remainders were shorter than 3 bp (Table 3). The 7 bp overlapping region "ATGATAA" (Fig 6B) has also been documented in several lepidopterans sequenced to date [82, 83].

The intergenic spacers of *Orthaga olivacea* Warre mitogenomes spread over fourteen regions and ranged in size from 1 to 52 bp with a total length of 134 bp. The longest intergenic spacer (52 bp) resided between *tRNA^Gln^* and *nad2*. The 20 bp intergenic spacer region located between *tRNA^Ser2^* and *nad1* contained the 'ATACTAA' motif. The 7 bp motif is considered to be a conserved structure found in most of the insect mtDNAs (Fig 6A).

## The A+T-rich region

The mitogenome of *Orthaga olivacea* Warre includes an A+T-rich region of 293 bp. This region showed the highest A+T content (93.86%), within the range reported of other lepidopterans (Table 4). Variation in intergenic length of noncoding regions particularly repeat sequences is responsible for most size variation in mitogenome. The control region is usually the largest noncoding part in the mitogenome [84, 85]. Several conserved structures found in other lepidopteran mitogenomes were also observed in the AT-rich region of *Orthaga olivacea* Warre, including the 'ATAGA' motif followed by a 17 bp poly-T stretch, and a microsatellite-like (AT)$_{13}$ reapeat [86, 87] (Fig 6C).

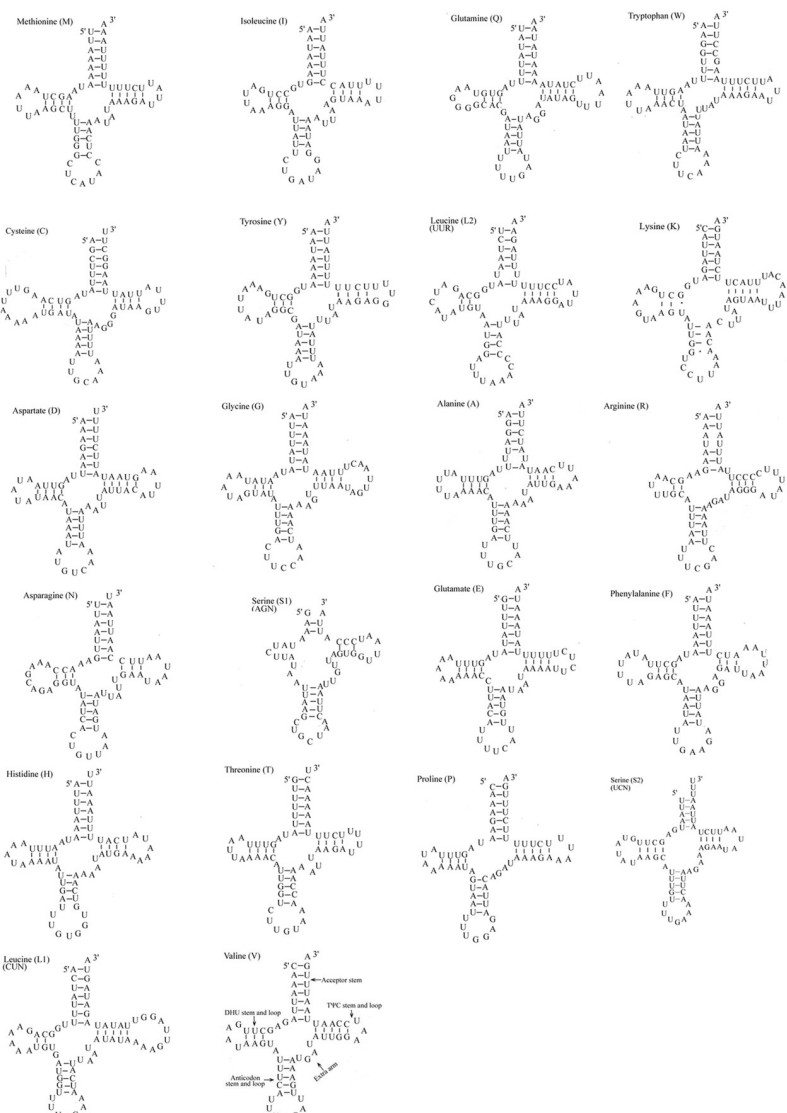

**Fig 5. Putative secondary structures of the 22 tRNA genes of the *Orthaga olivacea* Warre mitogenome.**

Above all, there are many remarkable characteristics in nucleotide composition. Compared with reported lepidopteran species, these characteristics include the structure of tRNAs and PCGs, A+T rich region and intergenic spacer region share similarities but also some differences. And these differences and similarities between them can be used as potential markers in phylogenetic analysis.

## Phylogenetic analysis

We reconstructed the phylogenetic relationships among seven lepidopteran superfamilies using Maximum Likelihood (ML) method based on concatenated nucleotide sequences of the 13 PCGs. Phylogenetic analysis revealed that different species from the same family clustered together (Fig 7). The complete nucleotide sequences of 59 species of Lepidoptera, represent 16

**Table 4. Composition and skewness in different Lepidopteran mitogenomes.**

| Species | Size (bp) | A% | G% | T% | C% | A+T % | ATskewness | GCskewness |
|---|---|---|---|---|---|---|---|---|
| **Whole genome** | | | | | | | | |
| **O. olivacea Warre** | **15174** | **37.83** | **8.23** | **41.18** | **12.75** | **79.02** | **−0.042** | **−0.215** |
| B. mori | 15643 | 43.05 | 7.32 | 38.27 | 11.36 | 81.32 | 0.051 | −0.216 |
| R. menciana | 15301 | 41.42 | 7.82 | 37.45 | 13.31 | 78.86 | 0.050 | −0.259 |
| M. sexta | 15516 | 40.67 | 7.46 | 41.11 | 10.76 | 81.79 | −0.005 | −0.181 |
| E. pseudoconspersa | 15461 | 40.42 | 7.61 | 39.51 | 12.46 | 79.93 | 0.011 | −0.241 |
| C. agnata | 15261 | 39.58 | 7.71 | 41.52 | 11.2 | 81.1 | −0.023 | −0.184 |
| A. cinerarium | 15722 | 41.51 | 7.80 | 39.32 | 11.37 | 80.83 | 0.027 | −0.186 |
| D. saccharalis | 15490 | 40.87 | 7.42 | 39.15 | 12.56 | 80.02 | 0.021 | −0.258 |
| C. medinalis | 15388 | 40.36 | 7.45 | 41.58 | 10.61 | 81.94 | −0.030 | −0.175 |
| 1P. distinctalis | 15354 | 41.04 | 7.49 | 41.22 | 10.24 | 82.27 | −0.002 | −0.155 |
| L. haraldusalis | 15213 | 40.47 | 7.66 | 41.04 | 10.83 | 81.52 | −0.007 | −0.172 |
| G. mellonella | 15320 | 38.62 | 7.47 | 41.80 | 12.11 | 80.42 | −0.039 | −0.237 |
| S. lechriaspis | 15368 | 39.86 | 7.63 | 41.34 | 11.17 | 81.19 | −0.018 | −0.188 |
| A. ilia | 15,242 | 39.77 | 7.75 | 40.68 | 11.80 | 80.45 | −0.011 | −0.207 |
| P. xylostella | 16179 | 40.66 | 7.68 | 40.22 | 10.82 | 80.89 | 0.005 | −0.170 |
| **PCG** | | | | | | | | |
| **O. olivacea Warre** | **11147** | **37.12** | **9.11** | **40.24** | **13.53** | **77.36** | **−0.040** | **−0.195** |
| B. mori | 11177 | 42.92 | 8.17 | 36.66 | 12.26 | 79.57 | 0.079 | −0.200 |
| R. menciana | 11225 | 40.97 | 8.58 | 36.12 | 14.33 | 77.1 | 0.063 | −0.251 |
| M. sexta | 11185 | 40.41 | 8.23 | 39.88 | 11.48 | 80.30 | 0.007 | -0.165 |
| E. pseudoconspersa | 11187 | 3969 | 8.43 | 38.3 | 13.58 | 77.99 | 0.017 | −0.233 |
| C. agnata | 11238 | 39.12 | 8.37 | 40.79 | 11.72 | 79.91 | −0.020 | −0.166 |
| A. cinerarium | 11227 | 40.63 | 8.78 | 38.19 | 12.39 | 78.83 | 0.031 | −0.171 |
| D. saccharalis | 11206 | 40.34 | 8.27 | 37.55 | 13.83 | 77.90 | 0.036 | −0.252 |
| C. medinalis | 11210 | 39.88 | 8.15 | 40.69 | 11.28 | 80.56 | −0.010 | −0.161 |
| P. distinctalis | 11189 | 40.54 | 8.12 | 40.53 | 10.81 | 81.07 | 0 | −0.142 |
| L. haraldusalis | 11193 | 39.88 | 8.47 | 40.16 | 11.49 | 80.04 | −0.003 | −0.151 |
| G. mellonella | 11196 | 38.03 | 8.20 | 40.84 | 12.92 | 78.88 | −0.036 | −0.224 |
| S. lechriaspis | 11256 | 39.30 | 8.35 | 40.41 | 11.93 | 79.72 | −0.014 | −0.177 |
| A. ilia | 11,148 | 39.41 | 8.41 | 39.49 | 12.69 | 78.89 | −0.001 | −0.203 |
| P. xylostella | 11049 | 40.47 | 8.82 | 38.85 | 11.86 | 79.32 | 0.020 | −0.147 |
| **tRNA** | | | | | | | | |
| **O. olivacea Warre** | **1452** | **39.461** | **8.26** | **40.70** | **11.57** | **80.17** | **−0.015** | **−0.167** |
| B. mori | 1468 | 42.10 | 7.90 | 39.31 | 10.69 | 81.40 | 0.034 | −0.150 |
| R. menciana | 1485 | 41.08 | 8.08 | 39.93 | 10.91 | 81.01 | 0.014 | −0.149 |
| M. sexta | 1554 | 40.99 | 7.92 | 41.06 | 10.04 | 82.05 | −0.001 | −0.118 |
| E. pseudoconspersa | 1466 | 41.41 | 8.19 | 40.18 | 10.23 | 81.58 | 0.015 | −0.111 |
| C. agnata | 1477 | 41.23 | 8.19 | 40.22 | 10.36 | 81.45 | 0.012 | −0.117 |
| A. cinerarium | 1483 | 42.01 | 8.02 | 39.45 | 10.52 | 81.46 | 0.031 | −0.135 |
| D. saccharalis | 1478 | 41.81 | 7.713 | 40.32 | 10.15 | 82.14 | 0.018 | −0.136 |
| C. medinalis | 1475 | 41.29 | 8.00 | 40.81 | 9.90 | 82.10 | 0.006 | −0.106 |
| P. distinctalis | 1536 | 42.19 | 8.14 | 39.78 | 9.9 | 81.97 | 0.029 | −0.098 |
| L. haraldusalis | 1451 | 41.08 | 7.86 | 41.42 | 9.65 | 82.49 | −0.004 | −0.102 |
| G. mellonella | 1489 | 40.09 | 8.06 | 40.90 | 10.95 | 80.51 | −0.010 | −0.152 |
| S. lechriaspis | 1450 | 40.97 | 8.00 | 40.90 | 10.14 | 81.86 | 0.001 | −0.118 |
| A. ilia | 1433 | 40.61 | 8.30 | 40.96 | 10.12 | 81.58 | −0.004 | −0.099 |

(*Continued*)

**Table 4.** (Continued)

| Species | Size (bp) | A% | G% | T% | C% | A+T % | ATskewness | GCskewness |
|---|---|---|---|---|---|---|---|---|
| *P. xylostella* | 1468 | 42.51 | 8.17 | 38.83 | 10.49 | 81.34 | 0.045 | −0.124 |
| **rRNA** | | | | | | | | |
| *O. olivacea* **Warre** | **2169** | **39.65** | **4.84** | **44.35** | **11.16** | **84.00** | **−0.056** | **−0.389** |
| *B. mori* | 2158 | 43.74 | 4.59 | 41.06 | 10.61 | 84.80 | 0.032 | −0.396 |
| *R. menciana* | 2147 | 43.04 | 4.84 | 40.71 | 11.41 | 83.74 | 0.028 | −0.404 |
| *M. sexta* | 2168 | 41.37 | 4.84 | 44.05 | 9.73 | 85.42 | −0.031 | −0.335 |
| *E. pseudoconspersa* | 2225 | 42.56 | 4.54 | 42.11 | 10.79 | 84.67 | 0.005 | −0.408 |
| *C. agnata* | 2112 | 40.01 | 5.07 | 44.65 | 10.27 | 84.66 | −0.055 | −0.339 |
| *A.cinerarium* | 2179 | 43.97 | 4.77 | 41.17 | 10.10 | 85.13 | 0.033 | −0.358 |
| *D. saccharalis* | 2193 | 41.45 | 6.84 | 43.59 | 10.17 | 85.04 | −0.025 | −0.360 |
| *C. medinalis* | 2170 | 41.47 | 5.02 | 43.87 | 9.63 | 85.35 | −0.028 | −0.314 |
| *P. distinctalis* | 2174 | 41.31 | 5.34 | 44.02 | 9.34 | 85.33 | −0.032 | −0.272 |
| *L. haraldusalis* | 2121 | 42.20 | 4.67 | 43.33 | 9.81 | 85.53 | −0.013 | −0.355 |
| *G. mellonella* | 2143 | 40.18 | 4.95 | 44.19 | 10.69 | 84.37 | −0.048 | −0.367 |
| *S. lechriaspis* | 2160 | 41.71 | 4.95 | 43.84 | 9.49 | 85.56 | −0.025 | −0.314 |
| *A. ilia* | 2109 | 40.11 | 4.98 | 44.86 | 10.05 | 84.97 | −0.056 | −0.337 |
| *P. xylostella* | 2162 | 41.44 | 4.90 | 43.94 | 9.71 | 85.38 | −0.029 | −0.329 |
| **A+T-rich region** | | | | | | | | |
| *O. olivacea* **Warre** | **293** | **44.03** | **2.73** | **49.83** | **3.41** | **93.86** | **−0.062** | **−0.111** |
| *B. mori* | 449 | 44.69 | 1.60 | 50.70 | 3.00 | 95.39 | −0.063 | −0.304 |
| *R. menciana* | 357 | 43.7 | 3.36 | 47.34 | 5.6 | 91.04 | −0.040 | −0.250 |
| *M. sexta* | 324 | 45.06 | 1.54 | 50.31 | 3.09 | 95.37 | −0.005 | −0.335 |
| *E. pseudoconspersa* | 388 | 43.56 | 2.32 | 50.26 | 3.87 | 93.81 | −0.071 | −0.250 |
| *C. agnata* | 334 | 46.71 | 1.5 | 46.71 | 5.09 | 93.41 | 0.000 | −0.545 |
| *A. cinerarium* | 625 | 47.20 | 1.92 | 48.64 | 2.24 | 95.84 | −0.015 | −0.077 |
| *D. saccharalis* | 335 | 43.28 | 0.60 | 51.64 | 4.48 | 94.93 | −0.088 | −0.765 |
| *C. medinalis* | 339 | 42.48 | 0.88 | 53.39 | 3.24 | 95.87 | −0.114 | −0.571 |
| *P. distinctalis* | 349 | 46.13 | 1.15 | 49 | 3.72 | 95.13 | −0.030 | −0.528 |
| *L. haraldusalis* | 310 | 45.81 | 0.97 | 50.32 | 2.90 | 96.13 | −0.047 | −0.499 |
| *G. mellonella* | 350 | 44.29 | 0.29 | 52.86 | 2.57 | 97.14 | −0.088 | −0.8 |
| *S. lechriaspis* | 441 | 40.36 | 2.49 | 52.38 | 4.76 | 92.74 | −0.130 | −0.313 |
| *A. ilia* | 403 | 42.93 | 3.23 | 49.63 | 4.22 | 92.56 | −0.072 | −0.133 |
| *P. xylostella* | 1081 | 37.74 | 2.50 | 45.42 | 5.09 | 83.16 | −0.092 | −0.341 |

families (*Bombycidae, Saturniidae, Sphingidae, Lymantriidae, Erebidae, Notodontidae, Noctuidae, Nolidae, Geometridae, Crambidae, Pyralidae, Tortricidae, Papilionidae, Nymphalidae, Plutellidae,* and *Lyonetiidae*) were downloaded from GenBank to reconstruct phylogenetic relationships among them. The species *Orthaga olivacea* Warre belonging to the superfamily Pyralidae, and the relationship were closer with *Hypsopygia regina* than that with *Galleria mellonella* and *Corcyra cephalonica*. Phylogenetic analyses showed that Pyraloidea is clustered with other superfamilies including Bombycoidea, Geometroidea, Noctuoidea, Papilionoidea, Tortricoidea, and Yponomeutoidea. Of these Bombycoidea and Geometroidea were sister groups, and the relationgship of them were closer than Noctuoidea in ML analysis (Fig 7). In the present study, the relationships at superfamily level are consistent with prior studies of lepidopteran phylogeny [88–90]. Previous classifications of Pyralidae species were mostly based

**A**

| | |
|---|---|
| *Orthaga olivacea* **Warre (Lepidoptera: Pyralidiae)** | TT ATACTAA AAAATAATCAAA |
| *Lisata haraldusalis* (Lepidoptera: Pyralidae) | TT ATACTAA ATAAAATTTACTTT |
| *Tyspanodes hypsalis* (Lepidoptera: Crambidae ) | ATACTAA AAATAATAAA |
| *Bombyx mandarina* (Lepidoptera:Bombyciade) | TTATTCA ATACTAA AAATATTACAA |
| *Antheraea pernyi* (Lepidoptera:Saturniidae) | ATACTAA AAATAATTCAAT |
| *Ctenoplusia agnata* (Lepidoptera: Noctuidae) | ATACTAA AAATAAATCAAT |
| *Apocheima cinerarium* (Lepidoptera: Geometridae) | ATACTAA AAAAATTATAATT |
| *Spilonota lechriaspis* (Lepidoptera:Tortricidae) | ATACTAA AAAAAATATAT |
| *Luehdorfia taibai* (Lepidoptera:Papilionidae) | ATACTAA AAATATTTA |
| *Plutella xylostella* (Lepidoptera: Plutellidae) | ATACTAA ATTTAAATAA |

**B**

ATP8

| | |
|---|---|
| *Orthaga olivacea* **Warre (Pyralididae)** | tRNA-Asp−3854 • • • TGAAA**ATGATAA**CTAAC • • • 5478−COX3 |
| *Lisata haraldusalis* (Pyralididae) | tRNA-Asp−3887 • • • TGAAA**ATGATAA**CTAAT • • • 4725−COX3 |
| *Tyspanodes hypsalis* (Crambidae ) | tRNA-Asp−3925 • • • TGAAA**ATGATAA**GAAAT • • • 4751−COX3 |
| *Bombyx mandarina* (Bombyciade) | tRNA-Asp−14289 • • • TGAAA**ATGATAA**CAAAC • • • 15121−COX3 |
| *Antheraea pernyi* (Saturniidae) | tRNA-Asp−3943 • • • TGAAA**ATGATAA**GTAAT • • • 4780−COX3 |
| *Ctenoplusia agnata* (Noctuidae) | tRNA-Asp−3901 • • • TGAAA**ATGATAA**GAAAT • • • 4733−COX3 |
| *Apocheima cinerarium* (Geometridae) | tRNA-Asp−3921 • • • TGAAA**ATGATAA**GAAAT • • • 4756−COX3 |
| *Spilonota lechriaspis* (Tortricidae) | tRNA-Asp−3888 • • • TGAAA**ATGATAA**GAAAT • • • 4720−COX3 |
| *Luehdorfia taibai* (Papilionidae) | tRNA-Asp−3902 • • • TGAAA**ATGATAA**GAAAT • • • 4725−COX3 |
| *Plutella xylostella* (Plutellidae) | tRNA-Asp−3890 • • • TGAAA**ATGATAA**GAAAT • • • 4728−COX3 |

ATP6

**C**

rrnS-14,881-
ATGTAAAATAAATAAC**ATAGA**ATTTTTTTTTTTTTTTTTTATATTAAAATATTTATTATAAATTATTAAATTT
TAAATATTTACTTTTCTTTTTTTCCCTAATATTAATTTGAAAATTAATAGTTATATTGATTTAAGTAATATT
CATTTAAATAAAAATATATTAATATTATTATTAATTAATAGGTAAATTTAATTAATTATTAATAATATTAATA
TATTAAATTATTTAATATATATATATATATATATATATATTTAAACCATTTCTAATAAATTTTATATATAAATAAT
A-15,174-trnM

**Fig 6. Conserved sequence across the Lepidoptera order.** (A) Intergenic spacer region alignment between *trnS2* (UCN) and *ND1* of several Lepidopterans. The framework 'ATACTAA' motif is conserved across the Lepidoptera order. (B) Intergenic overlap region alignment between *ATP8* and *ATP*6 of several Lepidopterans. The bold 'ATGATAA' motif is the overlap region and it's conserved across the Lepidoptera order. (C) Features present in the A+T-rich region of *Orthaga olivacea* Warre. The sequence is shown in the reverse strand. The ATAGA motif is bolded. The poly-T stretch is underlined. The single microsatellite T/A repeat sequence are double underlined.

on morphology, of which numerous studies are regionally limited; therefore, the precise position of Pyralidae within the Pyraloidea remained unclear, more studies are needed on the complete mitochondrial genome of the diverse Pyraloidea species in order to understand the complexity of phylogenetic relationships.

## Conclusion

The newly accessible mitogenome of *Orthaga olivacea* Warre (Lepidoptera: Pyralidae) is 15,174 bp long, including 13 protein-coding genes (PCGs), two rRNA genes, 22 tRNA genes and an A+T-rich region. The arrangement of 13 PCGs is same to that of other sequenced lepidopterans. All PCGs of the mitogenome start with typical ATN codons, except for cytochrome c oxidase 1 (*cox1*) with the start codon CGA. The canonical termination codon (TAA or TAG) occurs in nine PCGs (TAA for *nad2*, *cox1*, *cox2*, *atp8*, *atp6*, *nad3*, *nad5*, *nad4L* and *nad6* genes), and the remainders PCGs were terminated with a single T or TA (a single T for *cox3* and *cytb* genes, TA for *nad4* and *nad1* genes). Phylogenetic analysis suggested that *Orthaga olivacea* Warre is more closely related to the *Lista haraldusalis*, and confirms that *Orthaga olivacea* Warre belongs to the family Pyralidae.

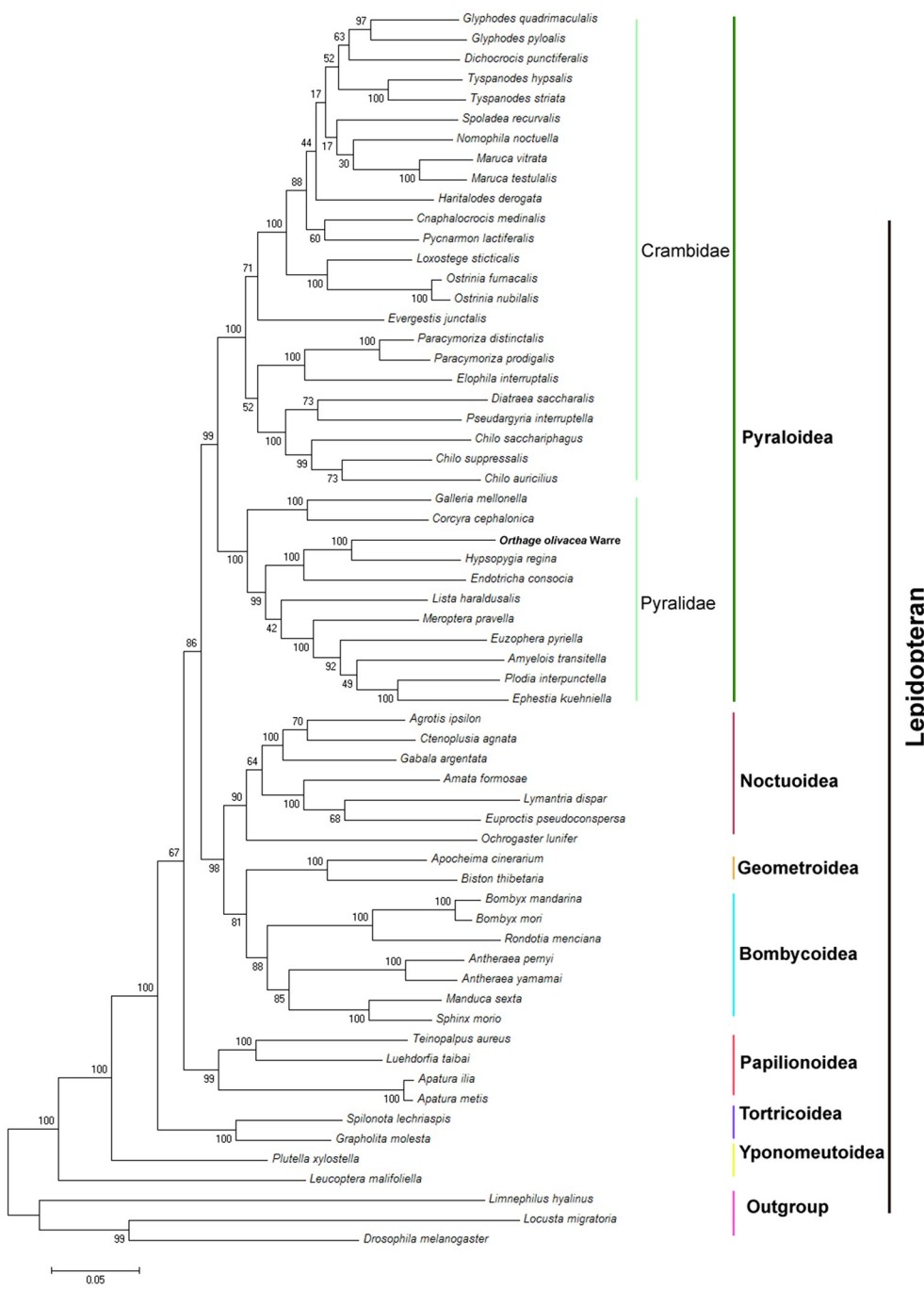

**Fig 7. Phylogenetic relationships tree among Lepidopteran insects.** The Maximum Likelihood method was used in the tree constructing. Bootstrap values (1000 repetitions) of the branches are indicated. *Limnephilus hyalinus* (NC_044710.1), *Drosophila incompta* (NC_025936) and *Locusta migratoria* (JN858212) were used as outgroups.

## Author Contributions

**Conceptualization:** Liangli Yang, Guoqing Wei.

**Data curation:** Yu Sun, Yuxuan Sun.

**Formal analysis:** Junjun Dai.

**Funding acquisition:** Guoqing Wei.

**Investigation:** Liangli Yang.

**Methodology:** Junjun Dai.

**Project administration:** Chaoliang Liu, Guoqing Wei.

**Resources:** Guoqing Wei.

**Software:** Liangli Yang.

**Supervision:** Baojian Zhu.

**Validation:** Qiuping Gao, Guozhen Yuan, Jiang Liu.

**Visualization:** Lei Wang, Cen Qian.

**Writing – original draft:** Liangli Yang.

**Writing – review & editing:** Junjun Dai, Guoqing Wei.

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
