## [Decision Letter · Decision Letter 0]

25 Oct 2019

PONE-D-19-24590

Characterization of the complete mitochondrial genome of Orthaga olivacea Warre (Lepidoptera Pyralididae) and comparison with other Lepidopteran insects

PLOS ONE

Dear Dr Wei,

Thank you for submitting your manuscript to PLOS ONE. After careful consideration, we feel that it has merit but does not fully meet PLOS ONE’s publication criteria as it currently stands. Therefore, we invite you to submit a revised version of the manuscript that addresses the points raised during the review process.

The manuscript has been assessed by three reviewers; their comments are available below.

The reviewers have raised major concerns that need attention in a revision. The reviewers note that the phylogenetic analysis needs to be substantially revised to add more species and to revisit the approach employed for the analyses. The reviewers also feel that the manuscript should provide a clearer outline of the study aims and research question addressed in the Introduction section and they request improvements to the written language.

Could you please carefully revise the manuscript to address the concerns raised by the reviewers?

We would appreciate receiving your revised manuscript by Dec 08 2019 11:59PM. To enhance the reproducibility of your results, we recommend that if applicable you deposit your laboratory protocols in protocols.io, where a protocol can be assigned its own identifier (DOI) such that it can be cited independently in the future. For instructions see: http://journals.plos.org/plosone/s/submission-guidelines#loc-laboratory-protocols

We look forward to receiving your revised manuscript.

Kind regards,

Iratxe Puebla

Senior Managing Editor, PLOS ONE

**Journal Requirements:**

2. We noticed you have some occurrence of overlapping text between the current submission and your previous publication(s) as following, which needs to be addressed:

- https://journals.plos.org/plosone/article?id=10.1371/journal.pone.0132951

-https://www.nature.com/articles/srep39153

-https://www.sciencedirect.com/science/article/pii/S0141813017324807?via%3Dihub

In your revision ensure you cite all your sources (including your own works), and quote or rephrase any duplicated text outside the methods section. Further consideration is dependent on these concerns being addressed.""

**Comments to the Author**

1. Is the manuscript technically sound, and do the data support the conclusions?

Reviewer #1: Partly

Reviewer #2: Yes

Reviewer #3: Yes

2. Has the statistical analysis been performed appropriately and rigorously? 

Reviewer #1: No

Reviewer #2: Yes

Reviewer #3: N/A

3. Have the authors made all data underlying the findings in their manuscript fully available?

Reviewer #1: Yes

Reviewer #2: Yes

Reviewer #3: Yes

4. Is the manuscript presented in an intelligible fashion and written in standard English?

Reviewer #1: Yes

Reviewer #2: Yes

Reviewer #3: Yes

5. Review Comments to the Author

Reviewer #1: In this manuscript, the authors sequence the complete mitochondrial genome of the moth, Orthaga olivacea. They then describe the annotation of this moth and use phylogenetic methods to compare it with the mitochondrial genomes of 34 other Lepidoptera. The writing is somewhat uneven—I have a number of suggestions for improvement, especially for the Abstract and Introduction in the Minor comments section at the end of this reviewer report. The annotation of the mitochondrial genome is acceptable, but I have a number of concerns especially about the phylogenetic analysis that will need to be addressed before this manuscript will be suitable for publication.

Major issues:

1. Pyraloidea taxon sampling. Line 62. “Considering the limited information of the mitochondrial sequences in Pyralidae, we sequenced the complete mitogenome of Orthaga olivacea, and compared it with other insect species, especially with the members of Pyralidae species.” This is a reasonable justification for sequencing the mitogenome of O. olivacea, but it is very curious that only 3 mitogenomes were included in the phylogenetic analysis and many of the other pyralid mitochondrial genomes that are available from Genbank were not included in these analyses including Corcyra cephalonica, Amyelois transitella, Plodia interpunctella (3 mitogenomes), Ephestia kuehniella (3 mitogenomes), Meroptera pravella, and Hypsopygia regina. Similarly, the sister-family to the Pyralidae includes an even larger number of species with sequenced mitochondrial genomes that were not included in the presented analyses including Paracymoriza prodigalis, Elophila interruptalis, Pseudargyria interruptella, Chilo auriculius, Chilo sacchariphagus, Evergestis junctalis, Nomophila noctuella, , Tryspandoes striata, Glyphodes quadrimaculalis, Spoladea recurvalis, Dichocrocis punctiferalis, Glyphodes pyloalis, Maruca vitrata, Maruca testulalis, Haritalodes derogate, Pycnarmon lactiferalis, Loxostege stricticalis, Endotricha consoci, Euzophera pyriella, Dichocrocis punctiferalis, and Cnaphalocrocis medinalis (3 mitogenomes). If one of the goals of the authors is to demonstrate that Orthaga olivacea belongs within the Pyralidae and to determine its closest relatives with sequences mitochondrial genomes, then they need to repeat their phylogenetic analysis after supplementing their current data set with all of these additional species. (Also note that Lista haraldusalis is misspelled in Fig. 7 and in other locations in the manuscript. Also, Family Pyralidae (and probably also Family Crambidae) should be indicated in Fig. 7.)

2. The authors employ 2 non-Lepidoptera outgroup species: Drosophila yakuba, a fly (Order Diptera) and Locusta migratoria, a grasshopper (Order Orthoptera), but the authors do not include any representatives of the insect Order most closely related to the Lepidoptera, the caddisflies (Order Trichoptera). There are at least 17 complete mitochondrial genomes representing several caddisfly Families available through Genbank (Al-Baeity et al. 2019). To root the Lepidopteran tree properly caddisflies sequences MUST also be included in the phylogenetic analyses.

Minor issues:

Line 2. Title: In modern usage, the lepidopteran family is usually called Pyralidae, not “Pyralididae”. Change here and throughout manuscript.

Line 18. Abstract: Suggested reword with greater specificity “Orthaga olivacea Ware (Lepidoptera Pyralidae) is an important agricultural pest of camphor trees (Cinnamomum camphora).”

Line 19. Suggested reword “To further supplement the known genome-level…”

Line 20. Suggested reword “…other species of Lepidoptera.”

Lines 31-31. Suggested reword “Phylogenetic analysis suggested that among sequenced lepidopteran mitochondrial genomes, Orthaga olivacea Warre was most closely related…”

Line 38. Suggested reword “…(mtDNA) is a circular molecule range in size from 14 to 19 kb…”

Line 42. Suggested reword “…A+T-rich region, the largest noncoding…”

Line 43-47. Suggested reword to remove repetition “Whole mitochondrial genomes are a useful data source for several research areas, such as evolutionary genomics (9, 10), comparative molecular evolution (11, 12), phylogeography (13), and population genetics (14).”

Line 50-51. Suggested reword to remove extraneous information “…over 25,000 species and some pyralids are important agricultural pests…”

Line 53-53. Suggested reword “Despite their diversity and great importance as pests of agricultural and forestry plants, they are also valuable for pollinating plants of economic importance. Most species in the family Pyralidae do not yet have sequenced mitogenomes.”

Line 58. “remove” should be “removal”

Line 59-60. “However, overlapping generations and irregularity of abundance in the field from May to October make it very difficult to control.”

Line 72. Suggested reword “…the camphor trees on the campus of…”

Line 94. “…insert DNA were sequenced at least three times…” Query: was sequencing of the inserts done in both directions? If yes, please specify in the text.

Lines 99-100. Suggested reword “…under the accession number MN078362.”

Line 125-126. “…mitogenome sizes documented for other sequenced lepidopterans which range from 14,534 bp in Ostrinia nublilalis (incomplete)…” Since the sequencing of the mitochondrial genome of O. nublilalis is incomplete, it is inappropriate and incorrect to use this sequence to estimate the minimum mitochondrial genome size in the Lepidoptera. This data point should be replaced with the smallest completely sequenced mitochondrial genome from the Lepidoptera.

Lines 136-137. Suggested reword “In addition, the GC skew…”

Line 146. Table 4. I’m not sure that this table is necessary and perhaps should be removed.

Line 166. Suggested reword “…observed in most other lepidopteran mitogenomes and are…”

Line 177-179. Suggested reword “…for instance, L. haraldusalis, G. mellonella, B. mori, B. thibetaria, and L. malifoliella species all lack GCT codons, while G. mollesta lacks CGT codons.”

Line 246-248. “The species Orthaga olivacea…” This sentence should be revised based on the updated phylogenetic analysis after adding the taxa I suggested in the major revisions section above.

Lines 252-253. Suggested reword “…constituent with prior studies of lepidopteran phylogeny.”

References:

Al-Baeity, H., Allard, L.S., Arreza, L., et al. (2019) The complete mitochondrial genome of the North American pale summer sedge caddisfly Limnephilus hyalinus (Insecta: Trichoptera: Limnephilidae). Mitochondrial DNA Part B 4: 413-415.

Reviewer #2: The manuscript mainly determined the complete mitochondrial genome of Orthaga olivacea Warre (Lepidoptera Pyralididae) and compare the mtDNA with other Lepidopteran insects. The English is acceptable. The literature cited is appropriate and draws on numerous comparative examples of similar research. Overall structure is of good quality and the raw data complete. The paper touches on the pertinent theoretical ideas proposed by earlier researchers. Overall, this manuscript is interesting, the description of the methods is complete and sound, and worthy to be published in “PLoS ONE” after minor modified.

1. the tables would be “three line”.

2. the literature 34 (line 353) was not complete.

3. “Warre” (ects.) in the figures would not be italic.

4. correct others, for examples, line 194 “TRNAs” (tRNAs ?); line 201 “The rNAs” (The rRNAs ?), ect.

Reviewer #3: In the manuscript, the mitogenome of Orthaga olivacea was determined and comparison with other lepidopteran sequences were also analyzed. The results of the study are valuable for the readers interested in the comparative mitogenome and phylogeny of Pyralididae. These results are informative and useful. I suggest this article can be published in this journal. However, the manuscript needs to be improved before acceptance for publication.

1. Introduction: the authors should provide clearly the study aim and scientific questions. It includes a description of the importance of the research and the study and reviews most of the previous literature. However, the authors have omitted a few studies of relevance and these should be included,

2. It is not clear from the manuscript that the collected Orthaga olivacea samples were verified astruly belong to the said species. It is suggested for the author to delimit the detailed morphological characters of the species to confirm.

3. Based on the dataset of 13 concatenated protein sequences, the authors reconstructed the phylogeny of Lepidoptera using MEGA with the Maximum Likelihood method. It is more persuasive and popular to carry out such analysis with RAxML method.

4. There are some errors in grammar and syntax throughout the text of the manuscript, the English writing should be further improved.

6. PLOS authors have the option to publish the peer review history of their article (what does this mean?). If published, this will include your full peer review and any attached files.

Reviewer #1: No

Reviewer #2: Yes: Ping You

Reviewer #3: No

---

## [Author Response · Author response to Decision Letter 0]

7 Dec 2019

Responds to the reviewer’s comments:

Reviewer #1: 

Major issues:

1. Response to comment: (Pyraloidea taxon sampling. Line 62. “Considering the limited information of the mitochondrial sequences in Pyralidae, we sequenced the complete mitogenome of Orthaga olivacea, and compared it with other insect species, especially with the members of Pyralidae species.” This is a reasonable justification for sequencing the mitogenome of O. olivacea, but it is very curious that only 3 mitogenomes were included in the phylogenetic analysis and many of the other pyralid mitochondrial genomes that are available from Genbank were not included in these analyses including Corcyra cephalonica, Amyelois transitella, Plodia interpunctella (3 mitogenomes), Ephestia kuehniella (3 mitogenomes), Meroptera pravella, and Hypsopygia regina. Similarly, the sister-family to the Pyralidae includes an even larger number of species with sequenced mitochondrial genomes that were not included in the presented analyses including Paracymoriza prodigalis, Elophila interruptalis, Pseudargyria interruptella, Chilo auriculius, Chilo sacchariphagus, Evergestis junctalis, Nomophila noctuella, , Tryspandoes striata, Glyphodes quadrimaculalis, Spoladea recurvalis, Dichocrocis punctiferalis, Glyphodes pyloalis, Maruca vitrata, Maruca testulalis, Haritalodes derogate, Pycnarmon lactiferalis, Loxostege stricticalis, Endotricha consoci, Euzophera pyriella, Dichocrocis punctiferalis, and Cnaphalocrocis medinalis (3 mitogenomes). If one of the goals of the authors is to demonstrate that Orthaga olivacea belongs within the Pyralidae and to determine its closest relatives with sequences mitochondrial genomes, then they need to repeat their phylogenetic analysis after supplementing their current data set with all of these additional species. (Also note that Lista haraldusalis is misspelled in Fig. 7 and in other locations in the manuscript. Also, Family Pyralidae (and probably also Family Crambidae) should be indicated in Fig. 7.))

Response: We are very sorry for our omission that it is inadequate to the goals to demonstrate that Orthaga olivacea belongs within the Pyralidae and to determine its closest relatives with sequences mitochondrial genomes with only 3 pyralid mitochondrial genomes were included in the phylogenetic analysis. According reviewer’s suggestion, we have repeated our phylogenetic analysis after supplementing our current data set with all of these additional species. And we have corrected the misspelled of Lista haraldusalis in Fig. 7 and in other locations in the manuscript. Also, Family Pyralidae (and probably also Family Crambidae) was indicated in Fig. 7.

2. Response to comment: (The authors employ 2 non-Lepidoptera outgroup species: Drosophila yakuba, a fly (Order Diptera) and Locusta migratoria, a grasshopper (Order Orthoptera), but the authors do not include any representatives of the insect Order most closely related to the Lepidoptera, the caddisflies (Order Trichoptera). There are at least 17 complete mitochondrial genomes representing several caddisfly Families available through Genbank (Al-Baeity et al. 2019). To root the Lepidopteran tree properly caddisflies sequences MUST also be included in the phylogenetic analyses.)

Response: Considering the Reviewer’s suggestion, we have included the caddisflies sequences of Limnephilus hyalinus in the phylogenetic analyses as outgroup.

Minor issues:

1. Response to comment: (Line 2. Title: In modern usage, the lepidopteran family is usually called Pyralidae, not “Pyralididae”. Change here and throughout manuscript.)

Response: We are very sorry for our Negligence of the use of “Pyralididae”, and we have corrected it to “Pyralidae” throughout manuscript.

2. Response to comment: (Line 18. Abstract: Suggested reword with greater specificity “Orthaga olivacea Ware (Lepidoptera Pyralidae) is an important agricultural pest of camphor trees (Cinnamomum camphora).”)

Response: According reviewer’s suggestion, we have reworded with greater specificity.

3. Response to comment: (Line 19. Suggested reword “To further supplement the known genome-level…”)

Response: According reviewer’s suggestion, we have reworded it in the target location.

4. Response to comment: (Line 20. Suggested reword “…other species of Lepidoptera.”)

Response: According reviewer’s suggestion, we have reworded it in the target location.

5. Response to comment: (Lines 31-31. Suggested reword “Phylogenetic analysis suggested that among sequenced lepidopteran mitochondrial genomes, Orthaga olivacea Warre was most closely related…”)

Response: According reviewer’s suggestion, we have reworded it in the target location.

6. Response to comment: (Line 38. Suggested reword “…(mtDNA) is a circular molecule range in size from 14 to 19 kb…”)

Response: According reviewer’s suggestion, we have reworded it in the target location.

7. Response to comment: (Line 42. Suggested reword “…A+T-rich region, the largest noncoding…”)

Response: According reviewer’s suggestion, we have reworded it in the target location.

8. Response to comment: (Line 43-47. Suggested reword to remove repetition “Whole mitochondrial genomes are a useful data source for several research areas, such as evolutionary genomics (9, 10), comparative molecular evolution (11, 12), phylogeography (13), and population genetics (14).)

Response: According reviewer’s suggestion, we have removed repetition in the target location.

9. Response to comment: (Line 50-51. Suggested reword to remove extraneous information “…over 25,000 species and some pyralids are important agricultural pests…”)

Response: According reviewer’s suggestion, we have removed extraneous information in the target location.

10. Response to comment: (Line 53-53. Suggested reword “Despite their diversity and great importance as pests of agricultural and forestry plants, they are also valuable for pollinating plants of economic importance. Most species in the family Pyralidae do not yet have sequenced mitogenomes.”)

Response: According reviewer’s suggestion, we have reworded it in the target location.

11. Response to comment: (Line 58. “remove” should be “removal”)

Response: According reviewer’s suggestion, we have corrected “remove” to “removal”.

12. Response to comment: (Line 59-60. “However, overlapping generations and irregularity of abundance in the field from May to October make it very difficult to control.”)

Response: According reviewer’s suggestion, we have reworded it in the target location.

13. Response to comment: (Line 72. Suggested reword “…the camphor trees on the campus of…”)

Response: According reviewer’s suggestion, we have reworded it in the target location.

14. Response to comment: (Line 94. “…insert DNA were sequenced at least three times…” Query: was sequencing of the inserts done in both directions? If yes, please specify in the text.)

Response: Yes, the sequencing of the inserts was done in both directins. According reviewer’s suggestion, we have specified in the text.

15. Response to comment: (Lines 99-100. Suggested reword “…under the accession number MN078362.”)

Response: According reviewer’s suggestion, we have reworded it in the target location.

16. Response to comment: (Line 125-126. “…mitogenome sizes documented for other sequenced lepidopterans which range from 14,534 bp in Ostrinia nublilalis (incomplete)…” Since the sequencing of the mitochondrial genome of O. nublilalis is incomplete, it is inappropriate and incorrect to use this sequence to estimate the minimum mitochondrial genome size in the Lepidoptera. This data point should be replaced with the smallest completely sequenced mitochondrial genome from the Lepidoptera.)

Response: Thank you for pointing out the error. We have re-searched NCBI, and found that the sequencing of the mitochondrial genome of Ostrinia nublilalis is complete with 14,535 bp. And maybe Ostrinia nublilalis is the smallest completely sequenced mitochondrial genome from the Lepidoptera. we have corrected it in the target location.

17. Response to comment: (Lines 136-137. Suggested reword “In addition, the GC skew…”)

Response: According reviewer’s suggestion, we have reworded it in the target location.

18. Response to comment: (Line 146. Table 4. I’m not sure that this table is necessary and perhaps should be removed.)

Response: Thank you for your suggestion, but we think by base preference and compared it with other species in table 4, can better understand this mitochondrial genome. Therefore, we chose to keep table 4 in the manuscript.

19. Response to comment: (Line 166. Suggested reword “…observed in most other lepidopteran mitogenomes and are…”)

Response: According reviewer’s suggestion, we have reworded it in the target location.

20. Response to comment: (Line 177-179. Suggested reword “…for instance, L. haraldusalis, G. mellonella, B. mori, B. thibetaria, and L. malifoliella species all lack GCT codons, while G. mollesta lacks CGT codons.”)

Response: According reviewer’s suggestion, we have reworded it in the target location.

21. Response to comment: (Line 246-248. “The species Orthaga olivacea…” This sentence should be revised based on the updated phylogenetic analysis after adding the taxa I suggested in the major revisions section above.)

Response: According reviewer’s suggestion, we have revised this sentence based on the updated phylogenetic analysis.

22. Response to comment: (Lines 252-253. Suggested reword “…constituent with prior studies of lepidopteran phylogeny.”)

Response: According reviewer’s suggestion, we have reworded it in the target location.

Reviewer #2: 

1. Response to comment: (the tables would be “three line”.)

Response: According reviewer’s suggestion, we have changed all tables into “three line” forms.

2. Response to comment: (the literature 34 (line 353) was not complete.)

Response: According reviewer’s suggestion, we have reworded it in the target location.

3. Response to comment: (“Warre” (ects.) in the figures would not be italic.)

Response: We are very sorry for our error application of the italic of “Warre” (ects.) in the figures, we have corrected it in all figures.

4. Response to comment: (“correct others, for examples, line 194 “TRNAs” (tRNAs ?); line 201 “The rNAs” (The rRNAs ?), ect.)

Response: We are very sorry for our incorrect in spelling, and we have already corrected them in the text and marked in color.

Reviewer #3: 

1. Response to comment: (Introduction: the authors should provide clearly the study aim and scientific questions. It includes a description of the importance of the research and the study and reviews most of the previous literature. However, the authors have omitted a few studies of relevance and these should be included,)

Response: According reviewer’s suggestion, we have reworded the Introduction section to provide a clearer outline of the study aims and research question and marked in color.

2. Response to comment: (It is not clear from the manuscript that the collected Orthaga olivacea samples were verified as truly belongs to the said species. It is suggested for the author to delimit the detailed morphological characters of the species to confirm.)

Response: According reviewer’s suggestion, we have added detailed description of the morphological characteristics of the species' larvae in the materials and methods section to better delimit the species.

3. Response to comment: (Based on the dataset of 13 concatenated protein sequences, the authors reconstructed the phylogeny of Lepidoptera using MEGA with the Maximum Likelihood method. It is more persuasive and popular to carry out such analysis with RAxML method.)

Response: We are very sorry that we didn’t reconstruct the phylogeny of Lepidoptera using the RAxML method as you suggested. Because we think the RAxML method is an alternative solution in phylogeny, we found that using MEGA method to analyze the phylogeny of Lepidoptera is also popular. For example, in the study of Cerura menciana (Dai et. al., 2015), Biston marginata (Zheng et al., 2018) and Ctenoptilum vasava (Hao et. al., 2012) in Lepidoptera, they used MEGA to reconstruct the evolutionary relationship of Lepidoptera with the Maximum Likelihood method, and also got a better evolutionary relationship tree. In this study, based on the analysis of the original evolutionary relationship, we added another 25 species of Pyralidae and finally got a better evolutionary relationship of Lepidoptera. So we think that the MEGA method can also be used to construct the evolutionary tree based on Lepidoptera mitochondria.

4. Response to comment: (There are some errors in grammar and syntax throughout the text of the manuscript, the English writing should be further improved.)

Response: We are very sorry for our incorrect in grammar and syntax, and we have already corrected them in the text and marked in color.

We tried our best to improve the manuscript and made some changes in the manuscript. These changes will not influence the content and framework of the paper. And here we did not list the changes but marked in revised manuscript. We appreciate for Editors/Reviewers’ warm work earnestly, and hope that the corrections will meet with approval. Once again, thank you very much for your comments and suggestions.

Yours Sincerely

Guoqing Wei

---

## [Editor Report · Decision Letter 1]

16 Dec 2019

PONE-D-19-24590R1

Characterization of the complete mitochondrial genome of Orthaga olivacea Warre (Lepidoptera Pyralidae) and comparison with other Lepidopteran insects

PLOS ONE

Dear Dr Wei,

Thank you for submitting your manuscript to PLOS ONE. After careful consideration, we feel that it has merit but does not fully meet PLOS ONE’s publication criteria as it currently stands. Therefore, we invite you to submit a revised version of the manuscript that addresses the points raised during the review process.

We would appreciate receiving your revised manuscript by Jan 30 2020 11:59PM. To enhance the reproducibility of your results, we recommend that if applicable you deposit your laboratory protocols in protocols.io, where a protocol can be assigned its own identifier (DOI) such that it can be cited independently in the future. For instructions see: http://journals.plos.org/plosone/s/submission-guidelines#loc-laboratory-protocols

We look forward to receiving your revised manuscript.

Kind regards,

Jeffrey M. Marcus

Academic Editor

PLOS ONE

Additional Editor Comments (if provided):

Greetings. After receiving the first round of reviewer comments responding to your initial submission, PLOS ONE has asked me to change my role from Reviewer (I was Reviewer #1) to Guest Academic Editor to guide you through the remainder of the peer review process.

I have read your revision and overall, I am very pleased with how you have responded to the reviewer comments. However, there are a few remaining items that you will need to address before your manuscript can be considered acceptable for publication in PLOS ONE. They are listed below. Please make theses additional necessary changes and resubmit your work for final consideration by the journal.

1. Line 62. Delete entire sentence beginning with "What'more considering the limited..." It is unnecessary.

2. Table 2 includes a column of references. These table citations are not in the same format as the in-text citations in the rest of the manuscript and some of these references do not appear in the reference section at the end of the manuscript. Please correct the formatting, and ensure that all of the references listed in Table 2 also appear in the reference section.

3. The reference for the Meroptera pravella mitochondrial genome in Table 2 is listed as "Consortium et al. (2017)". This is properly referenced as "Living Prairie Consortium (2017)".

4. Fig. 7. The vertical line associated with the label "Pyraloidea" should extend from Glyphodes quadrimaculalis to Ephestia kuehniella in this figure.

---

## [Author Response · Author response to Decision Letter 1]

27 Dec 2019

Dear Editor:

Thank you very much for your letter and the comments concerning our manuscript entitled “Characterization of the complete mitochondrial genome of Orthaga olivacea Warre (Lepidoptera Pyralidae) and comparison with other Lepidopteran insects” (ID: PONE-D-19-24590). Those comments are all valuable and very helpful for revising our paper to meet the acceptable criterion for publication in PLOS ONE. We have studied comments carefully and have made corrections which we hope to meet with approval. Revised portions are marked in color in the manuscript. The main corrections in the paper and the responds to the comments are as flowing:

Responds to the reviewer’s comments:

1. Response to comment: (Line 62. Delete entire sentence beginning with "What'more considering the limited..." It is unnecessary.)

Response: According to the reviewer’s suggestion, we have deleted it in the target location.

2. Response to comment: (Table 2 includes a column of references. These table citations are not in the same format as the in-text citations in the rest of the manuscript and some of these references do not appear in the reference section at the end of the manuscript. Please correct the formatting, and ensure that all of the references listed in Table 2 also appear in the reference section.)

Response: We are very sorry for our negligence of the citations format and omission of some references in Table 2, and we have corrected the formatting in Table 2 and increased the omissive references in the reference section.

3. Response to comment: (The reference for the Meroptera pravella mitochondrial genome in Table 2 is listed as "Consortium et al. (2017)". This is properly referenced as "Living Prairie Consortium (2017)".)

Response: Considering the second suggestion, we have modified the reference formats in table 2.

4. Response to comment: (Fig. 7. The vertical line associated with the label "Pyraloidea" should extend from Glyphodes quadrimaculalis to Ephestia kuehniella in this figure.)

Response: According the suggestion, we modified the vertical line associated with the label "Pyraloidea" and make sure it is extended from Glyphodes quadrimaculalis to Ephestia kuehniella in fig. 7.

We tried our best to improve the manuscript and made some changes in the manuscript. These changes will not influence the content and framework of the paper. And here we did not list the changes but marked in revised manuscript. We appreciate for Editors/Reviewers’ warm work earnestly, and hope that the corrections will meet with approval. Once again, thank you very much for your comments and suggestions.

Yours Sincerely

Guoqing Wei

---

## [Editor Report · Decision Letter 2]

31 Dec 2019

Characterization of the complete mitochondrial genome of Orthaga olivacea Warre (Lepidoptera Pyralidae) and comparison with other Lepidopteran insects

PONE-D-19-24590R2

Dear Dr. Wei,

We are pleased to inform you that your manuscript has been judged scientifically suitable for publication and will be formally accepted for publication once it complies with all outstanding technical requirements.

With kind regards,

Jeffrey M. Marcus

Guest Editor

PLOS ONE

Additional Editor Comments (optional):

Thank you for responding to my recommendations for revision. I am now prepared to recommend acceptance of this manuscript at PLOS ONE.
---

## [Editor Report · Acceptance letter]

24 Feb 2020

PONE-D-19-24590R2 

Characterization of the complete mitochondrial genome of *Orthaga olivacea* Warre (Lepidoptera Pyralidae) and comparison with other Lepidopteran insects 

Dear Dr. Wei:

I am pleased to inform you that your manuscript has been deemed suitable for publication in PLOS ONE. Congratulations! Your manuscript is now with our production department. 

With kind regards,

on behalf of

Dr. Jeffrey M. Marcus 

Guest Editor

PLOS ONE